# A Comparative Evaluation of Multiparametric Magnetic Resonance Imaging and Micro-Ultrasound for the Detection of Clinically Significant Prostate Cancer in Patients with Prior Negative Biopsies

**DOI:** 10.3390/diagnostics14050525

**Published:** 2024-03-01

**Authors:** Edoardo Beatrici, Nicola Frego, Giuseppe Chiarelli, Federica Sordelli, Stefano Mancon, Cesare Saitta, Fabio De Carne, Giuseppe Garofano, Paola Arena, Pier Paolo Avolio, Andrea Gobbo, Alessandro Uleri, Roberto Contieri, Marco Paciotti, Massimo Lazzeri, Rodolfo Hurle, Paolo Casale, Nicolò Maria Buffi, Giovanni Lughezzani

**Affiliations:** 1Department of Biomedical Sciences, Humanitas University, 20072 Pieve Emanuele, MI, Italy; edoardo.beatrici@humanitas.it (E.B.); nicola.frego@gmail.com (N.F.); giuseppe.chiarelli@humanitas.it (G.C.); federica.sordelli@humanitas.it (F.S.); stefano.mancon@humanitas.it (S.M.); cesare.saitta@humanitas.it (C.S.); fabio.decarne@humanitas.it (F.D.C.); giuseppe.garofano@humanitas.it (G.G.); paola.arena@humanitas.it (P.A.); pierpaolo.avolio@humanitas.it (P.P.A.); andrea.gobbo@humanitas.it (A.G.); alessandro.uleri@humanitas.it (A.U.); roberto.contieri@humanitas.it (R.C.); nicolo.buffi@hunimed.eu (N.M.B.); 2Department of Urology, IRCCS Humanitas Research Hospital, 20089 Rozzano, MI, Italy; marco.paciotti@humanitas.it (M.P.); massimo.lazzeri@humanitas.it (M.L.); rodolfo.hurle@humanitas.it (R.H.); paolo.casale@humanitas.it (P.C.)

**Keywords:** diagnosis, imaging, micro-ultrasound, multiparametric MRI, prostate cancer, prostate biopsy

## Abstract

Background: The diagnostic process for prostate cancer after a negative biopsy is challenging. This study compares the diagnostic accuracy of micro-ultrasound (mUS) with multiparametric magnetic resonance imaging (mpMRI) for such cases. Methods: A retrospective cohort study was performed, targeting men with previous negative biopsies and using mUS and mpMRI to detect prostate cancer and clinically significant prostate cancer (csPCa). Results: In our cohort of 1397 men, 304 had a history of negative biopsies. mUS was more sensitive than mpMRI, with better predictive value for negative results. Importantly, mUS was significantly associated with csPCa detection (adjusted odds ratio [aOR]: 6.58; 95% confidence interval [CI]: 1.15–37.8; *p* = 0.035). Conclusions: mUS may be preferable for diagnosing prostate cancer in previously biopsy-negative patients. However, the retrospective design of this study at a single institution suggests that further research across multiple centers is warranted.

## 1. Introduction

Prostate cancer is the most frequently diagnosed malignancy among men, with an estimated 300,000 new cases projected in the United States for 2024 [1]. A pivotal advancement in the process of diagnosing prostate cancer is the incorporation of multiparametric magnetic resonance imaging (mpMRI) combining targeted and systematic biopsies together, now worldwide recommended worldwide as the standard diagnostic approach to detecting prostate cancer, particularly for identifying incidences of high-grade disease [2,3,4]. Despite its widespread adoption, mpMRI encounters limitations, including restricted accessibility resulting in prolonged waiting periods, considerable associated costs, and the complexity of interpretation necessitating highly specialized radiologists with extensive expertise [3,5,6]. Additionally, variability in the methodology, such as the use of MRI machines with different magnetic strengths (e.g., 1.5 Tesla vs. 3 Tesla) and the option of an endorectal coil, can affect the results [7,8]. Furthermore, there is notable inter-reader reliability (IRR) among radiologists, which can lead to variations in diagnostic outcomes [9,10,11].

In this context, micro-ultrasound (mUS) (Exact Imaging, Markham, ON, Canada) emerges as an innovative technology. Operating at an elevated frequency of 29 MHz and with a resolution of 70 microns, mUS offers a potential alternative to mpMRI [12,13,14]. Its advantages are manifold: immediate availability, reduced costs, and the capability for urologists to perform and interpret the examination in real time. Empirical evidence has already substantiated the high sensitivity and specificity of mUS in the detection of prostate cancer and particularly clinically significant prostate cancer (csPCa) in biopsy-naïve patients [15,16,17,18,19,20]. The utility of identifying csPCa, considered a lesion of the International Society of Urological Pathology Grade Group (GG) ≥ 2, has been demonstrated to significantly predict cancer-specific mortality, reinforcing the importance of advanced diagnostic and prognostic tools in personalized clinical decision making [21,22].

Despite advancements in prostate cancer diagnostics, a particular challenge persists in managing patients with a history of negative biopsies and persistent clinical suspicious of prostate cancer [23,24]. For this subset, mpMRI has been the cornerstone of further diagnostic investigations [18,25]. However, the evidence base concerning the efficacy of mpMRI in this group, especially in comparison to biopsy-naïve patients, reveals certain limitations [26,27]. These patients generally exhibit a lower risk of prostate cancer diagnosis compared to those without prior biopsies, even when presenting with lesions of similar Prostate Imaging Reporting and Data System (PI-RADS) scores. Some researchers, acknowledging this knowledge gap, have suggested the adoption of specialized predictive models or a higher PI-RADS cut-off of 4 for recommending biopsies in these patients [28]. Such approaches underscore the evolving nature of diagnostic strategies in prostate cancer, particularly in complex cases involving patients with prior negative biopsy results. In this nuanced scenario, the potential role of mUS demands attention. Although mUS has demonstrated high sensitivity and specificity in detecting prostate cancer in biopsy-naïve patients, its effectiveness and utility in the context of patients with previous negative biopsies have not yet been thoroughly investigated and compared to mpMRI.

Therefore, the primary objective of the current study was to evaluate and compare the diagnostic accuracy of mUS and mpMRI in detecting prostate cancer and csPCa within a cohort of men with a history of negative prostate biopsy and presenting with clinical indications suggestive of prostate cancer. By investigating mUS’s diagnostic accuracy against mpMRI in this specific context, our research elucidates the potential of mUS to enhance diagnostic precision and accessibility, offering a novel insight into prostate cancer detection strategies for clinically complex cases.

## 2. Material and Methods

### 2.1. Data Source

In the current study, a dataset reporting data on mUS examinations, originating solely from the internal archives of our healthcare institution, was retrospectively analyzed. Ethical oversight was rigorously enforced throughout the investigative process. The experimental protocol was subjected to a thorough review process and received formal authorization from the institution’s Institutional Review Board (Protocol ICH 003 v1.0, approved on 27 September 2017, study number 2004) in strict compliance with the ethical guidelines outlined in the World Medical Association’s Declaration of Helsinki pertaining to human subjects in medical research [29].

### 2.2. Study Population

This study embarked on a retrospective analysis of our institution’s internal patient registry, focusing on individuals presenting with clinical indications suggestive of prostate cancer. This was evidenced by clinical examinations and prostate-specific antigen (PSA) levels. The selected patients underwent both mUS and mpMRI as part of their comprehensive diagnostic evaluation, specifically targeting those who had experienced at least one prior negative prostate biopsy.

In this study, male participants aged 18 years or older with a documented history of at least one prior negative prostate biopsy and no prior diagnosis of prostate cancer were included. When laboratory and clinical data indicated the necessity for further diagnostic evaluation, the men underwent both mpMRI and mUS as part of an enhanced prostate cancer screening protocol. The diagnostic imaging assessments were conducted within three months of each other.

The criteria for proceeding to an additional prostate biopsy were based on a decision-making process shared with the patient based on the clinical examination, the PSA value, the DRE, and the detection of lesions suggestive of prostate cancer by either imaging modality [30]. Lesions were categorized as suspicious if they presented with a PI-RADS score of 3 or higher for mpMRI and a Prostate Risk Identification using micro-ultrasound (PRI-MUS) score of 3 or higher for mUS. Patients exhibiting such suspicious lesions identified by either mpMRI or mUS were subjected to targeted biopsies, complemented by a minimum of 6 systematic cores, conforming to the European Association of Urology (EAU) guidelines for patients with previously negative prostate biopsies.

In instances where mpMRI and mUS identified different lesions suspected of harboring prostate cancer, at least 2 targeted biopsies were executed on both identified lesions. Importantly, to maintain the integrity of the study, the radiologist interpreting the mpMRI and the urologist conducting the mUS examination were blinded to each other’s findings.

### 2.3. Main Outcome Variable

The primary outcome variable was the presence of prostate cancer in targeted biopsies performed based on the mpMRI and mUS findings. The presence of csPCa was identified as a secondary outcome. Clinically significant cases were defined as those with an International Society of Urological Pathology (ISUP) Grade Group (GG) cancer grading of 2 or higher. This classification strategy was employed to distinguish between less aggressive prostate cancers (ISUP GG1), which may warrant a range of management options starting from expectant management (active surveillance and watchful waiting), and more aggressive forms (ISUP GG ≥ 3) that require immediate diagnosis, comprehensive treatment, and rigorous intervention strategies [31]. Prostate biopsy procedures were executed via either a transrectal or transperineal approach, conforming to the EAU guidelines pertinent to each patient’s assessment time. Starting in 2022 and thanks to the availability of the micro-ultrasound FusionVu system, mUS was adopted as the standard ultrasound probe for all biopsies, thanks to advancements enabling efficient mUS-mpMRI fusion biopsies.

### 2.4. Main Predictor Variable

The main predictor variable in our analyses was the type of imaging tool (mpMRI or mUS) that identified lesions suspicious of harboring prostate cancer.

### 2.5. Covariates

Age at the time of prostate biopsy was treated as a continuous variable. The last PSA value available at the last urological examination before performing the prostate biopsy was treated as a categorical variable and categorized as follows: <10 ng/mL; 10–20 ng/mL; >20 ng/mL. DRE was performed at urological discretion at the last urological examination, before performing imaging examinations, and was categorized as follows: negative, positive, and not performed. Prostate volume based on the measures obtained at mpMRI was categorized as follows: <40 mL; 40–80 mL; >80 mL. Prior surgery for benign prostatic obstruction (BPO) was dichotomized as follows: no, yes, and missing. The prostate biopsy approach was categorized as follows: transrectal, transperineal, combined (transrectal and transperineal), or missing (when not reported).

### 2.6. Statistical Analysis

Categorical variables were quantified using frequencies and proportions, while continuous variables, particularly those not adhering to a normal distribution, were expressed through median values and interquartile ranges. The efficacy of both imaging methodologies in detecting csPCa was assessed using the Pearson chi-squared test, which facilitated the calculation of sensitivity, specificity, positive predictive value (PPV), and negative predictive value (NPV).

Further analysis entailed complete case evaluations employing multivariable logistic regression models. These models were adjusted for the aforementioned covariates, thereby allowing for a nuanced assessment of the predictors in identifying prostate cancer and csPCa.

Statistical computations were executed using STATA software (Stata/SE 18.0 for Mac, Copyright 1985–2023 StataCorp LLC, StataCorp, 4905 Lakeway Drive, College Station, TX, USA). The statistical tests were bidirectional, adhering to a significance threshold of *p* < 0.05.

## 3. Results

### 3.1. Baseline Characteristics

In the time frame from October 2017 to February 2023, our institution observed a cohort of 1397 patients who underwent both mpMRI and mUS assessments as part of their prostate cancer screening protocol. Of these, 859 men (accounting for 61.5% of the total cohort) were classified as biopsy-naïve. The remaining 538 patients (38.5%) had a documented history of at least one previous prostate biopsy. Within this subgroup, 304 patients (56.5%) exhibited at least one negative biopsy result for prostate cancer in their antecedent assessments. These individuals were subsequently identified as the primary cohort of the current study.

The median age of the study population was 66 years (IQR: 61–71), while the median PSA value was 8.6 ng/mL (IQR: 5.8–11.5), and the median prostate volume was 52.6 mL (IQR: 49.5–77). Overall, 26/304 men (8.6%) underwent previous surgical treatment for BPO. Only 58/304 men (19.1%) had a positive DRE upon urological examination. Table 1 reports the population’s baseline characteristics.

At the imaging assessments, suspicious lesions (PI-RADS ≥ 3 or PRI-MUS ≥ 3) were detected in 201/304 (66.1%) patients and 221/304 (72.7%) patients through mpMRI and mUS, respectively.

### 3.2. mpMRI and mUS Results

Upon mpMRI examination, zero suspicious lesions were identified in 103/304 men (33.9%), one was identified in 150/304 men (49.3%), two were identified in 44/304 (14.5%), three were identified in 6/304 (2%), and four were identified in 1/304 (0.3%).

Upon mUS examination, zero suspicious lesions were identified in 84/304 men (27.6%), one suspicious lesion was identified in 151/304 men (49.7%), two suspicious lesions were identified in 65/304 (21.4%), and three suspicious lesions were identified in 4/304 (1.3%). Figure 1 and Figure 2 present mUS images from men with prior negative biopsies.

Overall, concordance between the two imaging methodologies in identifying suspicious lesions was registered in 193/304 (63.5%) cases.

### 3.3. Prostate Cancer and Clinically Significant Prostate Cancer Detection Rates

At biopsy, prostate cancer was diagnosed in 102 (33.6%) patients. Among the 102 men with a diagnosis of prostate cancer at prostate biopsy, only 57/102 (55.9%) men presented with csPCa. Among these, mpMRI identified lesions suspected of harboring csPCa (PI-RADS ≥ 3) in 44/57 (77.2%) of the cases, while mUS (PRIMUS ≥ 3) identified lesions suspected of harboring csPCa (PI-RADS ≥ 3) in 52/57 (91.2%) of the cases.

MicroUS demonstrated higher sensitivity (91.2% vs. 77.2%) and NPV (66.7% vs. 45.8%) compared to mpMRI in predicting csPCa among men with previous negative biopsies.

However, both imaging methodologies exhibited significantly low specificity (24.4% vs. 22.2% for mpMRI vs. mUS, respectively) and PPVs (59.8% vs. 56.4% for mpMRI vs. mUS, respectively)

Table 2 presents the baseline characteristics of patients, stratified by the diagnosis of clinically insignificant prostate cancer (ciPCa) versus csPCa.

### 3.4. Predictive Models for the Detection of Prostate Cancer and Clinically Significant Prostate Cancer

Upon conducting a univariable logistic regression, both a positive mpMRI result (OR 2.09; 95%CI 1.22–3.57; *p*-value = 0.007) and a positive mUS result (OR 2.94; 95%CI 1.58–5.46; *p*-value = 0.001) were identified as significant predictors for the presence of prostate cancer at biopsy. Nevertheless, when focusing on csPCa only, neither a positive mpMRI (OR 1.10; 95%CI 0.44–2.75; *p*-value 0.85) nor a positive mUS (OR 2.98; 95%CI 0.94–9.44; *p*-value = 0.07) resulted in a significantly association with csPCa in a univariable analysis.

In multivariable logistic regression models, both a positive mpMRI result (aOR 2.51; 95%CI 1.26–5.00; *p*-value = 0.009) and a positive mUS result (aOR 2.92; 95%CI 1.35–6.31; *p*-value = 0.007) were significantly associated with the presence of prostate cancer. When only considering csPCa, a positive mUS result was significantly associated with csPCa (aOR: 6.58; 95%CIs: 1.15–37.8; *p*-value = 0.035), whereas in the model with mpMRI, a positive mpMRI result was not associated with csPCa (aOR 2.25; 95% CI 0.63–8.09; *p*-value = 0.21). Table 3 and Table 4 report the results of the multivariable logistic regression models fitted to assess the association between a positive mpMRI (Table 3) or mUS (Table 4) result and the presence of prostate cancer and csPCa.

## 4. Discussion

Navigating the diagnosis of prostate cancer in patients with previous negative biopsies is complex, balancing the risk of undetected malignancies against over-reliance on invasive diagnostics [32]. This challenge is accentuated by limitations in conventional diagnostic tools and the diverse progression patterns of prostate cancers. It underscores the need for a meticulous, evidence-based approach in clinical decision making to discern malignancies accurately while minimizing unnecessary invasive procedures which may entail considerable adverse consequences [33,34]. In this context, the present study offers a critical analysis of two imaging modalities: mpMRI and mUS. We embarked on a comprehensive retrospective analysis to evaluate and compare the diagnostic efficacy of both imaging tools in detecting prostate cancer, specifically focusing on this challenging patient cohort. Our findings highlight the enhanced diagnostic capabilities of mUS, particularly its superior sensitivity, specificity, and NPV compared to mpMRI in identifying prostate cancer and particularly csPCa when dealing with men with previous negative biopsies. Furthermore, they suggest that a positive finding at mUS (PRIMUS ≥ 3) is strongly associated with the presence of csPCa. This insight is pivotal, as it points toward the potential of mUS to fill a crucial gap in the diagnostic process for these patients.

Prior negative prostate biopsies have been shown to correlate with a reduced rate of detection of prostate cancer, as categorized by PI-RADS scores, in comparison to biopsy-naïve individuals [28,35,36]. Nevertheless, conducting mpMRI prior to repeat biopsy procedures is crucial. In men with prior negative biopsies, mpMRI-guided targeted biopsies have consistently demonstrated a superior prostate cancer detection rate compared to systematic biopsy alone [37]. The extant literature indicates that the sensitivity of mpMRI in detecting prostate cancer in men with previous negative biopsies ranges between 78% and 100% [38]. Focusing on csPCa and excluding ISUP GG1 lesions, our study documented a sensitivity of 77.2% for mpMRI in identifying csPCa in this cohort [31,39] This contrasts with the markedly higher sensitivity of 91.2% observed for mUS in detecting csPCa. Additionally, mUS demonstrated superior NPV compared to mpMRI (66.7% vs. 45.8%, respectively), although both modalities exhibited notably low specificity (22.2% vs. 24.4% for mUS vs. mpMRI, respectively). PPV was slightly higher for mpMRI than for mUS (56.4% vs. 59.8%, respectively).

These diagnostic accuracy metrics indicate a trade-off: while high sensitivity and NPV favor the exclusion of significant disease, the low specificity and PPV reported for both imaging methodologies pose a risk of over-diagnosis and potentially superfluous prostate biopsies. This highlights the necessity of a balanced application of these imaging techniques and the careful interpretation of results within the broader clinical context. However, the diagnostic accuracy observed in our analysis concurs with the performance of mUS reported in other patient cohorts, including those who are biopsy-naïve, thereby reinforcing the consistency and reliability of mUS across diverse clinical settings [40,41]. In addition to this, the accuracy reported for mpMRI, although slightly lower, aligns with prior studies conducted on biopsy-naïve patients [42] This might highlight the very challenging management of patients with a history of a prior negative biopsy, where consolidated imaging methodologies might present limitations, therefore reinforcing the importance of a specific focus on this cohort to maximize their diagnostic pathway.

We posit that the availability of an accurate imaging tool like mUS, characterized by its high sensitivity and minimal requirement for specialist interpretation, is crucial in managing the challenging patient cohort of men with prior negative biopsies. Notably, it is hypothesized that mUS may have an abbreviated learning curve which, if confirmed by future research, could significantly enhance its practical utility in clinical settings. The superior accuracy of mUS compared to mpMRI, as evidenced in our study, suggests a potential shift in diagnostic strategy [43]. By reducing the reliance on mpMRI and complementing its routine use, mUS not only alleviates the burden on this resource-intensive modality but also potentially expedites access for biopsy-naïve patients.

In parallel, mUS enhances the diagnostic evaluation for patients with prior negative biopsies, a group often facing continuing uncertainty of prostate cancer risk. Notably, mUS has proven effective in discerning cases within the ambiguous realm of PI-RADS 3 lesions on mpMRI, adeptly minimizing the identification of ciPCa while reliably detecting csPCa [44]. Similarly, mUS has been proven to be as effective in the management of patients under active surveillance as event-triggered confirmatory biopsy [45]. This capability mirrors the potential benefits for patients with a history of negative biopsies, potentially reducing both the frequency of repeated biopsies and the likelihood of ciPCa detection.

Moreover, the repeatability of mUS as a relatively non-invasive procedure, and its cost-effectiveness compared to mpMRI, positions it as a valuable tool for ongoing patient monitoring, potentially circumventing the need for much more expensive and invasive diagnostic approaches while following up on suspicious lesions without performing biopsies. This consideration gains further importance in light of prostate cancer screening studies which demonstrate a very low mortality risk (approximately 1% at 12 years) following a negative prostate biopsy, questioning the extent to which new diagnostic interventions can enhance survival in this patient group [46,47,48].

Our study, while offering valuable insights into the diagnostic efficacy of mpMRI and mUS in detecting prostate cancer, particularly csPCa, in men with previous negative biopsies, is subject to certain limitations. The retrospective design, focused on a single healthcare institution’s internal data, potentially impacts the generalizability of our findings. This approach inherently carries the risk of selection bias and may not be fully representative of the broader population in different clinical settings. Additionally, the study’s findings have not undergone external validation, which is essential to substantiating the applicability of our results in other healthcare contexts. Another consideration is the potential for observer variability given that the interpretation of imaging findings was subject to individual radiologists’ and urologists’ discretion and expertise. Despite the blinding of the urologist and the radiologist to each other’s findings, this factor could have influenced the diagnostic outcomes. Furthermore, our study does not account for the long-term outcomes of patients, particularly those with csPCa, which limits our understanding of the clinical implications of our findings over time. Lastly, while we have made strides in integrating advanced technologies like the micro-ultrasound FusionVu system, the evolving nature of diagnostic technology means that our findings might need to be re-evaluated as newer methods and tools become available. These limitations underscore the need for cautious interpretation of our results and highlight the importance of further research in this domain, including prospective studies and multicenter collaborations, to validate and expand upon our findings.

## 5. Conclusions

Our study highlights the diagnostic advantage of mUS over mpMRI in detecting prostate cancer, particularly csPCa, in men with previous negative biopsies. With its higher sensitivity and NPV, mUS shows promise, especially in complex cases. This suggests a potential shift in prostate cancer diagnostics, balancing accuracy with cost-effectiveness and accessibility. Nonetheless, the limitations of this single-center retrospective study call for further validation through multicenter prospective research to confirm mUS’s role in wider clinical practice.

## Figures and Tables

**Figure 1 diagnostics-14-00525-f001:**
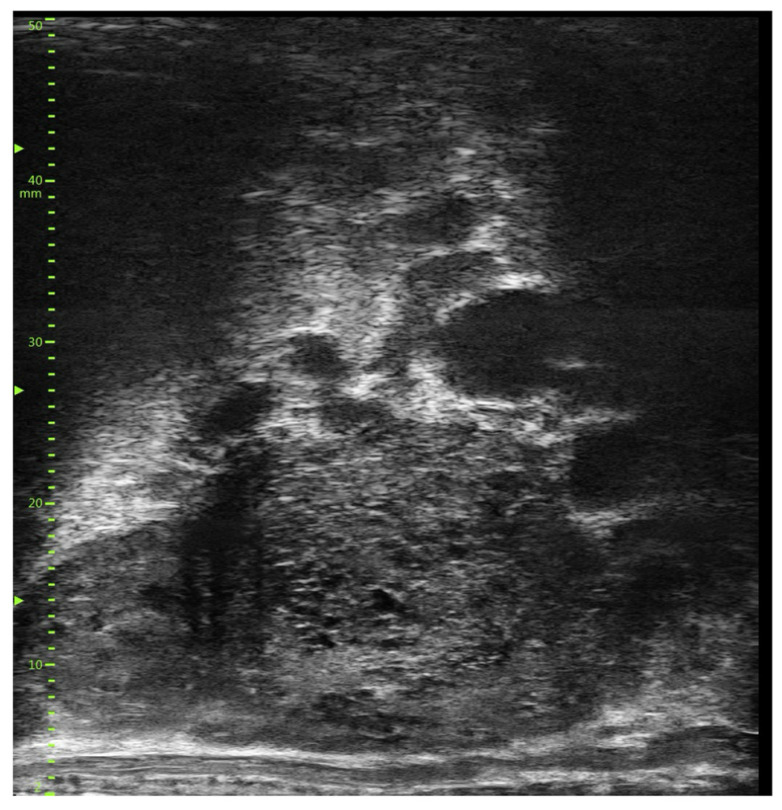
Micro-ultrasound image displaying a typical benign pattern with no lesions indicative of prostate cancer. The observed pattern is characterized by hyper-echogenic regions with ductal patches, categorized as PRI-MUS I, alongside small, orderly ductal structures, known as the ‘Swiss Cheese’ pattern, designated as PRI-MUS II.

**Figure 2 diagnostics-14-00525-f002:**
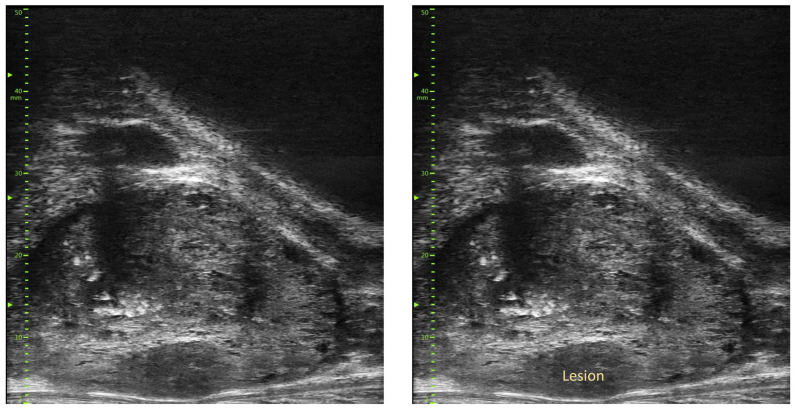
Micro-ultrasound image showing a smudgy (PRI-MUS IV) lesion in the peripheral zone of the prostate with an associated capsular bulging but no visible capsular breach.

**Table 1 diagnostics-14-00525-t001:** Baseline characteristics of patients with a prior negative prostate biopsy who underwent further diagnostic imaging evaluation through micro-ultrasound (mUS) and multiparametric magnetic resonance imaging (mpMRI) before undergoing prostate biopsy.

	Total
	*N* = 304
Age	66.0 (61.0–71.0)
Total PSA (ng/mL)	
<10	187 (61.5%)
10–20	94 (30.9%)
>20	23 (7.6%)
Prostate volume (mL)	
<40	58 (19.1%)
40–80	179 (58.9%)
>80	67 (22.0%)
Digital rectal examination	
Negative	206 (67.7%)
Positive	58 (19.1%)
Not performed	40 (13.2%)
Prior surgery for prostatic obstruction	
No	275 (90.5%)
Yes	26 (8.6%)
Missing	3 (1.0%)
Biopsy approach	
Transrectal biopsy	177 (58.2%)
Transperineal biopsy	76 (25.0%)
Combined (transrectal and transperineal) biopsy	49 (16.1%)
Missing	2 (0.7%)
PI-RADS score primary lesion:	
3	57 (18.8%)
4	104 (34.2%)
5	40 (13.2%)
No lesions identified	103 (33.9%)
PRI-MUS score primary lesion:	
3	27 (8.9%)
4	133 (43.8%)
5	60 (19.7%)
No lesions identified	84 (27.6%)

Data are presented as median (IQR) values for continuous measures, and *n* (%) for categorical measures. PI-RADS = Prostate Imaging Reporting and Data System; PRI-MUS = Prostate Risk Identification using Micro-Ultrasound; PSA = prostate-specific antigen.

**Table 2 diagnostics-14-00525-t002:** Baseline characteristics of patients with a prior negative prostate biopsy who underwent further diagnostic imaging evaluation through micro-ultrasound (mUS) and multiparametric magnetic resonance imaging (mpMRI), stratified according to the presence of clinically insignificant or clinically significant prostate cancer at biopsy.

	ciPCa	csPCa	Total	*p*-Value
	*N* = 45	*N* = 57	*N* = 102	
Age	66.5 (61.0–70.5)	70.0 (64.0–76.0)	68.0 (63.0–74.0)	0.007
Total PSA (ng/mL)				0.072
<10	30 (66.7%)	32 (56.1%)	62 (60.8%)	
10–20	14 (31.1%)	16 (28.1%)	30 (29.4%)	
>20	1 (2.2%)	9 (15.8%)	10 (9.8%)	
Prostate volume (mL)				0.053
<40	15 (33.3%)	18 (31.6%)	33 (32.4%)	
40–80	21 (46.7%)	36 (63.1%)	57 (55.9%)	
>80	9 (20.0%)	3 (5.3%)	12 (11.8%)	
Digital rectal examination				0.067
Negative	33 (73.3%)	30 (52.6%)	63 (61.8%)	
Positive	7 (15.6%)	20 (35.1%)	27 (26.5%)	
Not performed	5 (11.1%)	7 (12.3%)	12 (11.8%)	
Prior surgery for prostatic obstruction				0.30
No	41 (91.1%)	48 (84.2%)	89 (87.3%)	
Yes	4 (8.9%)	9 (15.8%)	13 (12.7%)	
Missing				0.15
Biopsy approach	22 (48.9%)	37 (64.9%)	59 (57.8%)	
Transrectal biopsy	10 (22.2%)	12 (21.1%)	22 (21.6%)	
Transperineal biopsy	13 (28.9%)	8 (14.0%)	21 (20.6%)	
PI-RADS score primary lesion:				0.035
3	4 (8.9%)	2 (3.5%)	6 (5.9%)	
4	23 (51.1%)	27 (47.4%)	50 (49.0%)	
5	8 (17.8%)	23 (40.4%)	31 (30.4%)	
No lesions identified	10 (22.2%)	5 (8.8%)	15 (14.7%)	
PRI-MUS score primary lesion:				0.12
3	9 (20.0%)	6 (10.5%)	15 (14.7%)	
4	21 (46.7%)	23 (40.4%)	44 (43.1%)	
5	4 (8.9%)	15 (26.3%)	19 (18.6%)	
No lesions identified	11 (24.4%)	13 (22.8%)	24 (23.5%)	

Data are presented as median (IQR) values for continuous measures and *n* (%) values for categorical measures. ciPCa = clinically insignificant prostate cancer; csPCa = clinically significant prostate cancer; PI-RADS = Prostate Imaging Reporting and Data System; PRI-MUS = Prostate Risk Identification using Micro-Ultrasound; PSA = prostate-specific antigen.

**Table 3 diagnostics-14-00525-t003:** Results of the multivariable logistic regression model fitted to assess the association between a positive multiparametric magnetic resonance imaging (mpMRI) result and the presence of prostate cancer and clinically significant prostate cancer. The model is adjusted for age, PSA value, prostate volume, digital rectal examination, prior surgery for prostatic obstruction, and biopsy approach.

	Prostate Cancer		Clinically Significant Prostate Cancer	
	Odds Ratio	95% CI	Odds Ratio	95% CI
Multiparametric Magnetic Resonance Imaging				
Negative	(Base)	(Base)	(Base)	(Base)
Positive	2.51 *	1.26–5.00	2.25	0.63–8.09
Age (per year increase)	1.07 **	1.03–1.12	1.08 *	1.01–1.16
PSA Categories (ng/mL)				
<10	1 (Base)	-	(Base)	(Base)
10–20	1.19	0.62–2.28	1.72	0.53–5.63
>20	1.35	0.45–4.02	18.18	0.93–354.55
Prostate Volume (cc)				
<40	(Base)	(Base)	(Base)	(Base)
40–80	0.39 *	0.19–0.80	1.14	0.36–3.62
>80	0.16 **	0.06–0.39	0.08 *	0.01–0.65
Digital Rectal Examination				
Negative	(Base)	(Base)	(Base)	(Base)
Positive	1.70	0.85–3.38	2.22	0.68–7.28
Prior Surgery for Prostatic Obstruction				
No	(Base)	(Base)	(Base)	(Base)
Yes	1.29	0.50–3.34	1.33	0.31–5.78
Biopsy Approach				
Transrectal Biopsy	(Base)	(Base)	(Base)	(Base)
Transperineal Biopsy	0.75	0.37–1.54	0.45	0.12–1.70
Combined Biopsy Approach (Transrectal and Transperineal)	1.02	0.44–2.35	0.22 *	0.05–0.98

* = *p*-value < 0.05; ** = *p*-value < 0.001.

**Table 4 diagnostics-14-00525-t004:** Results of the multivariable logistic regression model fitted to assess the association between a positive micro-ultrasound (mUS) examination result and the presence of prostate cancer and clinically significant prostate cancer. The model is adjusted for age, PSA value, prostate volume, digital rectal examination, prior surgery for prostatic obstruction, and biopsy approach.

	Prostate Cancer		Clinically Significant Prostate Cancer	
	Odds Ratio	95% CI	Odds Ratio	95% CI
Micro-Ultrasound Result				
Negative	(Base)	(Base)	(Base)	(Base)
Positive	2.92 *	1.35–6.32	6.58 *	1.15–37.78
Age (per year increase)	1.07 **	1.03–1.12	1.09 *	1.01–1.17
PSA Categories (ng/mL)				
<10 (Base)	(Base)	(Base)	(Base)	(Base)
10–20	1.07	0.56–2.05	1.90	0.56–6.45
>20	1.52	0.51–4.55	26.97 *	1.56–466.02
Prostate Volume (cc)				
<40	(Base)	(Base)	(Base)	(Base)
40–80	0.39 *	0.19–0.79	1.27	0.39–4.15
>80	0.19 **	0.08–0.47	0.11 *	0.01–0.83
Digital Rectal Examination				
Negative	(Base)	(Base)	(Base)	(Base)
Positive	1.32	0.67–2.63	1.92	0.56–6.58
Prior Surgery for Prostatic Obstruction				
No	(Base)	(Base)	(Base)	(Base)
Yes	1.49	0.56–3.93	1.48	0.32–6.85
Biopsy Approach				
Transrectal Biopsy	(Base)	(Base)	(Base)	(Base)
Transperineal Biopsy	1.08	0.53–2.19	0.57	0.16–2.06
Combined Biopsy Approach (Transrectal and Transperineal)	1.41	0.63–3.17	0.22	0.05–1.01

* = *p*-value < 0.05; ** = *p*-value < 0.001.

## Data Availability

The data presented in this study are available on request from the corresponding author. The data are not publicly available due to privacy concerns related to the individual data contained within the dataset.

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
