# Peer review of "A Comparative Evaluation of Multiparametric Magnetic Resonance Imaging and Micro-Ultrasound for the Detection of Clinically Significant Prostate Cancer in Patients with Prior Negative Biopsies"

_diagnostics, 2024, doi:10.3390/diagnostics14050525_

Round 1

Reviewer 1 Report

Comments and Suggestions for Authors

This is an interesting article on a current topic that supports the wider use of microultrasound in the detection of prostate cancer. It would be preferably for the authors to add to the article images of examples of positive microultrasound  and mpMRI findings of some of the patients included in the study.

Author Response

Author response:

We appreciate the reviewer's suggestion to include images of mUS and mpMRI findings from our study. Unfortunately, we could not obtain the necessary authorization to reproduce and include mpMRI images from our institution's Radiology department. As a result, we have focused on providing mUS images in the manuscript.

Despite this limitation, we believe that including mUS images offers significant value. Given the broader utilization and familiarity of mpMRI in the clinical setting, the mUS images we present may offer readers a novel perspective and potentially greater interest. These images underscore the diagnostic capabilities of mUS in detecting prostate cancer, especially in cases where patients had previously received negative biopsy results. By highlighting the utility of mUS, we aim to contribute to the growing interest in this imaging modality as a complementary tool in prostate cancer detection.

We trust that these adjustments and clarifications address the reviewer's concerns and enhance the manuscript's contribution to the field.

Changes, Results, Page 5: “Figures 1 and 2 present mUS images from men with prior negative biopsies.”

Reviewer 2 Report

Comments and Suggestions for Authors

The manuscript is presented in a highly structured manner. It includes a comprehensive review of the literature, balancing between foundational and recent studies. The data analysis itself raises several but critically important questions.

1.     The abstract contains too much technical information, which may hinder reader comprehension. Try simplifying sentence structures and highlighting key points to make your research essence more accessible. While presenting statistical results is important, too many numbers can dampen reader interest. Strive to include only essential figures and key findings. The abstract should be written clearly and straightforwardly. Avoid overly complex sentence constructions, and ensure each sentence is easily understandable. By addressing these comments, your abstract can become clearer and attract more reader attention.

2.     Please arrange the citation of literature sources according to the journal's requirements.

3.     You have a very extensive structured dataset. It could be used for solving tasks other than the ones you focused on. In the introduction, I missed a review of other authors' studies on the analysis of prostate cancer data. This would not only expand the manuscript's particularly brief introduction but also transition from a breadth review to a depth review of what is currently being done. I suggest incorporating relevant sources, which you don't necessarily have to rely on, and also hope that you will expand this proposed list yourself:

·       GibbonsM., et al. Prostate cancer lesion detection, volume quantification and high-grade cancer differentiation using cancer risk maps derived from multiparametric MRI with histopathology as the reference standard, Magnetic Resonance Imaging, Volume 99, 2023, pp. 48-57, https://doi.org/10.1016/j.mri.2023.01.006.

·       Milonas D, et al. Impact of Grade Groups on Prostate Cancer-Specific and Other-Cause Mortality: Competing Risk Analysis from a Large Single Institution Series. Cancers. 2021;13(8):1963. doi: 10.3390/cancers13081963.

4.     In the introduction, it is not enough to formulate only the objective, but the novelty of the work should also be emphasized.

5.     In the prognostic models presented in section 3.4, the proportions of response variable values are unclear. When related to the logistic analysis model in section 3.1 for the presence and absence of prostate cancer, was the ratio 304:1093 used? And for clinically significant prostate cancer, was the ratio 57:45 or 57:1340 used?

6.     Since you provide confidence interval limits in Table 3 and Table 4, p-values become less relevant. You can simply indicate, next to each Odds ratio value, a marker for whether the corresponding empirical significance level is less than 0.01, less than 0.05, etc. This would help reduce the space occupied by the tables.

Author Response

Reviewer 2:

The manuscript is presented in a highly structured manner. It includes a comprehensive review of the literature, balancing between foundational and recent studies. The data analysis itself raises several but critically important questions.

  1. The abstract contains too much technical information, which may hinder reader comprehension. Try simplifying sentence structures and highlighting key points to make your research essence more accessible. While presenting statistical results is important, too many numbers can dampen reader interest. Strive to include only essential figures and key findings. The abstract should be written clearly and straightforwardly. Avoid overly complex sentence constructions, and ensure each sentence is easily understandable. By addressing these comments, your abstract can become clearer and attract more reader attention.

Author response:

We appreciate the reviewer's constructive feedback. In response, we have carefully refined the abstract to enhance its clarity and improve its accessibility to our readers. We meticulously condensed complex information, ensuring that only essential figures and findings were highlighted. Our revised abstract now succinctly conveys the core findings, addressing the balance between detail and readability as suggested.

Changes, Abstract, Page 1: Abstract: “Background: The diagnostic process for prostate cancer after a negative biopsy is challenging. This study compares the diagnostic accuracy of Micro-Ultrasound (mUS) with Multiparametric Magnetic Resonance Imaging (mpMRI) for such cases. Methods: A retrospective cohort study was performed, targeting men with previous negative biopsies and using mUS and mpMRI to detect prostate cancer and clinically significant prostate cancer (csPCa). Results: In our cohort of 1,397 men, 304 had a history of negative biopsies. mUS was more sensitive than mpMRI, with better predictive value for negative results. Importantly, mUS was significantly associated with csPCa detection (adjusted Odds Ratio [aOR]: 6.58; 95% Confidence Interval [CI]: 1.15-37.8; p=0.035). Conclusions: mUS may be preferable for diagnosing prostate cancer in previously biopsy-negative patients. However, the retrospective design of this study at a single institution suggests that further research across multiple centers is warranted.”

  1. Please arrange the citation of literature sources according to the journal's requirements.

Author response:

We thank the reviewer for their guidance. The references have now been arranged according to the journal's specific requirements, as outlined in the Instructions for Authors.

  1. You have a very extensive structured dataset. It could be used for solving tasks other than the ones you focused on. In the introduction, I missed a review of other authors' studies on the analysis of prostate cancer data. This would not only expand the manuscript's particularly brief introduction but also transition from a breadth review to a depth review of what is currently being done. I suggest incorporating relevant sources, which you don't necessarily have to rely on, and also hope that you will expand this proposed list yourself:
  • GibbonsM., et al. Prostate cancer lesion detection, volume quantification and high-grade cancer differentiation using cancer risk maps derived from multiparametric MRI with histopathology as the reference standard, Magnetic Resonance Imaging, Volume 99, 2023, pp. 48-57, https://doi.org/10.1016/j.mri.2023.01.006.
  • Milonas D, et al. Impact of Grade Groups on Prostate Cancer-Specific and Other-Cause Mortality: Competing Risk Analysis from a Large Single Institution Series. Cancers. 2021;13(8):1963. doi: 10.3390/cancers13081963.

Author response:

We thank the reviewer for the valuable suggestion. In response, we have expanded the introduction to include a broader review of recent studies on prostate cancer data analysis. This enhancement not only extends our discussion on the current diagnostic and prognostic approaches but also situates our work within the wider context of ongoing research. Specifically, we have incorporated references to studies by Gibbons et al. and Milonas et al., among others, to enrich our literature review and underscore the diverse applications of our dataset beyond the initial scope of our research.

Changes: Introduction, Page 1, Line 34: “particularly for identifying high-grade diseases”

Changes: Introduction, Page 2, Lines 119-123: “The utility of identifying csPCa, considered a lesions of the International Society of Urological Pathology Grade Group (GG) ≥ 2, has been demonstrated to significantly predict cancer-specific mortality, reinforcing the importance of advanced diagnostic and prognostic tools in personalized clinical decision-making”

Changes, References added:

- Reference 4: Gibbons M, Simko JP, Carroll PR, Noworolski SM. Prostate cancer lesion detection, volume quantification and high-grade cancer differentiation using cancer risk maps derived from multiparametric MRI with histopathology as the reference standard. Magn Reson Imaging. 2023;99:48-57. doi:https://doi.org/10.1016/j.mri.2023.01.006

- Reference 20: Castillo T. JM, Arif M, Starmans MPA, et al. Classification of Clinically Significant Prostate Cancer on Multi-Parametric MRI: A Validation Study Comparing Deep Learning and Radiomics. Cancers (Basel). 2022;14(1). doi:10.3390/cancers14010012

- Reference 21: Milonas D, Ruzgas T, Venclovas Z, Jievaltas M, Joniau S. Impact of Grade Groups on Prostate Cancer-Specific and Other-Cause Mortality: Competing Risk Analysis from a Large Single Institution Series. Cancers (Basel). 2021;13(8). doi:10.3390/cancers13081963

- Reference 22: Epstein JI, Zelefsky MJ, Sjoberg DD, et al. A Contemporary Prostate Cancer Grading System: A Validated Alternative to the Gleason Score. Eur Urol. 2016;69(3):428-435. doi:https://doi.org/10.1016/j.eururo.2015.06.046

- Reference 25: Görtz M, Huber AK, Linz T, et al. Detection Rate of Prostate Cancer in Repeat Biopsy after an Initial Negative Magnetic Resonance Imaging/Ultrasound-Guided Biopsy. Diagnostics. 2023;13(10). doi:10.3390/diagnostics13101761

  1. In the introduction, it is not enough to formulate only the objective, but the novelty of the work should also be emphasized.

Author response:

We thank the reviewer for the comment. The introduction has been revised to underscore the study's novelty, specifically the assessment of mUS versus mpMRI for detecting prostate cancer in patients with prior negative biopsies. This revision clarifies the study's innovative contribution to diagnostic approaches in complex clinical cases, addressing the reviewer's comments for a more explicit emphasis on the research's uniqueness and potential impact.

Changes: Introduction, Page 2, Lines 143-146: “By investigating mUS's diagnostic accuracy against mpMRI in this specific context, our research elucidates the potential of mUS to enhance diagnostic precision and accessibility, offering a novel insight into prostate cancer detection strategies for clinically complex cases.”

  1. In the prognostic models presented in section 3.4, the proportions of response variable values are unclear. When related to the logistic analysis model in section 3.1 for the presence and absence of prostate cancer, was the ratio 304:1093 used? And for clinically significant prostate cancer, was the ratio 57:45 or 57:1340 used?

Author response:

We thank the reviewer for the insightful comment. In response, we wish to clarify that our analyses focused on a specific cohort of 304 patients, identified from a larger group of 1,397 individuals who underwent both mUS and mpMRI evaluations. This subgroup, selected based on their history of at least one negative prostate biopsy, serves as the basis for our logistic regression models discussed in sections 3.4. This clarification emphasizes our targeted approach and the clinical relevance of our findings to patients with prior negative biopsy results.

  1. Since you provide confidence interval limits in Table 3 and Table 4, p-values become less relevant. You can simply indicate, next to each Odds ratio value, a marker for whether the corresponding empirical significance level is less than 0.01, less than 0.05, etc. This would help reduce the space occupied by the tables.

            Author response:

We appreciate the reviewer's advice and have updated Tables 3 and 4 accordingly. We've removed the p-values and now use markers to indicate significance levels next to each OR, streamlining the presentation and improving readability.

Changes: Results, Pages 9-11, Tables 3 and 4

Round 2

Reviewer 2 Report

Comments and Suggestions for Authors

The manuscript has been significantly improved. It can be accepted for publication.